# Sex-Specific Perching: Monitoring of Artificial Plants Reveals Dynamic Female-Biased Perching Behavior in the Black Soldier Fly, *Hermetia illucens* (Diptera: Stratiomyidae)

**DOI:** 10.3390/insects15100770

**Published:** 2024-10-05

**Authors:** Noah B. Lemke, Lisa N. Rollison, Jeffery K. Tomberlin

**Affiliations:** 1Department of Entomology, Texas A&M University, 2475 HEEP, College Station, TX 77843, USA; 2Texas A&M AgriLife, 600 John Kimbrough Blvd #510, College Station, TX 77843, USA

**Keywords:** reproduction, breeding, cage design, habitat, energy, mass-rearing, perching, time-series

## Abstract

**Simple Summary:**

Female black soldier flies perch on artificial plants much more often than males, especially early in the day and early in their reproductive lives, times when males are competing with one another in aerial swarms.

**Abstract:**

Artificial perches are implemented by many companies that mass-rear the black soldier fly (BSF), to emulate a natural breeding environment or provide additional surface area for flies to rest; however, basic information about perching behavior is lacking. This experiment tested the effect of adding 0.00, 0.04, 0.26, or 0.34 m^2^ of surface area to 0.93 m^3^ cages, each supplied with 90 male and 90 female adults. Female thoraxes marked with acrylic paint, and the number of perching flies of each sex were recorded over 6 d. A time-series analysis revealed the following: (a) females utilized perches 1.42 times more often than males across two trials; (b) especially in the morning where the difference could be as high as 2.56 times as great; (c) this decreased to 0.20–1.57 times more females than males by 1600 h; and (d) this cyclical pattern repeated each day throughout the week with a decreasing female-bias, starting from 2.41-times more females on day 1, which fell to 0.88–1.98-times more females than males on day 6. These dynamics are likely due to the presence of male flies engaging in aerial contests near ultraviolet lamps required for mating, especially during the early hours and early adulthood, aligning with and expanding prior knowledge of black soldier fly mating behavior.

## 1. Introduction

The study of habitat structural complexity has been a longstanding and pervasive theme in ecology [1], dating back to the 1950s and before with Huffaker’s famous manipulation experiment in which he alternated the arrangement of an orange and rubber ball ‘universe’, affecting the population dynamics of herbivorous and predatory mites (Acaridae) [2]. For decades, researchers have endeavored to characterize both biotic and abiotic factors that impact animal behavior [3].

Take for example neotropical [4] detritivores, like the black soldier fly, *Hermetia illucens* (L. 1758) (Diptera: Stratiomyidae) [5]. Jenny’s State Equation [6], can be used as a formalized way to link spatially explicit factors in a single conceptual framework. The model posits that temperature [7] and humidity [8,9] increase the rate of nutrient cycling and soil formation, which is also dependent on several factors, such as the nutritional quality of the substrate [10,11], as well as larval competition [12].

The arrangement of objects in space is also linked de facto to selection and resource partitioning [13]. For this reason, it is likely that spatial complexity affects the efficiency of larvae as they process waste [14], such as through substrate depth [15], particle size [15], and the heterogeneity of non-digestible objects [16,17]. Although spatial factors, such as these, are understood to affect the growth and development of black soldier fly larvae, the effect of habitat conformation on reproductive behavior is less appreciated, even though it is known that alterations to the habitat (via domestication) can directly impact adult behavior and ultimately fitness [18].

Altering the habitat structure (whether through natural or artificial processes) can impact species behavior across multiple scales. Knowledge of this bottom-up process has been used in various applied contexts ranging from the design of artificial reefs [19] and urban greenspaces [20] to nature preserves [21]. As a specific example, in the mass-reared *Ceratitits capitata* (Weidemann, 1824) (Diptera: Tephritidae), modifying enclosures with slats [22] was shown to increase the number of lekking sites [23,24,25,26], which enhanced reproductive performance by reducing male-aggression [27].

Yet, for the similarly lekking black soldier fly [28,29], little is known about how adults explicitly interact with the structure of their habitat in any context. Besides a singular report [29], in which the authors describe males (with a 91.9% (n = 109) male-to-female sex ratio) perching on Kudzu *Pueraria montana* (Loer) (Merr.) [29], nothing is known about adult behavior in the wild.

Colony production methods for the black soldier fly initially describe providing artificial plants (50 cm globes, with leaves that resemble ivy measuring 3.8–7.6 cm) to flies that were misted twice-daily with an automatic water sprayer [30]. Because of this, or simply out of intuition, many industrial producers supply their cages with artificial plants, apparently to provide a resting area or increase the absolute surface area available relative to each fly [31]. However, the direct effects on fecundity or fitness have not been quantified, nor any on behavior (i.e., perching) or insect welfare [32,33] (e.g., stress-reduction [34]). If the inclusion of perches has no effect on fitness, this potentially represents an unneeded cost to cages used in a mass-rearing facility.

Perching behavior in insects is theorized to aid with mate-locating [35,36], thermoregulation [37], and the reduction of intra-specific competition [38] by allowing insects to vertically stratify. With regards to mate-locating behaviors, insects often perch as they wait for conspecifics to approach [29], meaning that theoretically, the inclusion of perches could facilitate reproduction by promoting natural behaviors.

Using McCoy and Bell’s conceptual model [39], habitat structural complexity was defined as the arrangement of objects in the environment that possesses a functional interaction and the organism being studied, at a suitable scale, but which are not explicitly necessary for survival or incompatible with life. Critically, McCoy and Bell’s definition [39] removes from consideration the oviposition substrate and possibly the cage itself, since the enclosure might inhibit processes that would happen if flies were uncaged. This perspective hence renders artificial plants the objects of focus. Such a functional interaction could be that plants or similar structures are utilized by males to first establish lekking sites [35,36], where females will visit to mate before returning to the oviposition site [5,25,29].

We then asked whether an increasing gradient of habitat structural complexity likewise positively affected reproductive outcomes in black soldier flies, hypothesizing that there would be an indirect effect on fertile egg production, but this effect will diminish if complexity is too great with respect to scale. Furthermore, on the basis that males were previously observed to associate with plants [29], we expected that perching behavior would be primarily associated with males.

## 2. Methods

All experimentation was conducted in the Forensic Laboratory for Investigative Entomological Sciences (FLIES) Facility (Texas A&M University, College Station, TX, USA). Two trials were conducted. Methods were modified from the following studies [40,41].

### 2.1. Experimental Design

In keeping with prior studies [40,41,42,43], twelve Insect-A-Hide Pop-Up Shelters (84 × 84 × 133 cm, L × W × H) (Lee Valley Tools Ltd., Ottawa, ON, Canada) were utilized as mating cages, and were arranged in a 2 × 6 array within room 9 of building 1043 of the FLIES Facility. Treatments (CTRL, LOW, MED, HIGH) were assigned randomly, such that each treatment category had three replicates. To each cage, a 50-watt light-emitting diode (LED) (HK SPR AGTECH Trading LTD, Hong Kong, China) was fixed at a height of 101 cm and distance 1 cm from the cage door [41]. During experimentation, lights were ran on a 12:12 L:D photoperiod (06:00–18:00 h) to approximate both the duration of natural light in a previous study [43] and in the tropics (personal observation) to which black soldier flies are native [44]. Because temperature and humidity were not controlled, these variables were recorded synchronously using a HOBO^®^ data logger MX1104 (Onset Computer Co., Bourne, MA, USA).

### 2.2. Artificial Plants

To construct artificial plants (see Appendix A), unfinished dowel rods (3.18 mm × 15 cm: dia × L) (EBOOT, Shenzhen City, Guangdong, China) were used as stems and attached to the underside of plastic artificial *Monstera deliciosa* Liebm. (Alismatales: Araceae) leaves (KUUQA Direct, Shenzhen City, Guangdong, China) using strips of 18 mm-wide masking tape (Scotch TM, St. Paul, MN, USA). *M. deliciosa*-shaped leaves were selected because it is a neotropical plant that is sympatric with *Hermetia illucens* in their native ranges. The artificial leaves measured 12.2 cm (small) or 17.5 cm (medium) in length from base to tip and 11.0 cm and 17.1 cm from margin to margin, respectively. Larger leaves from the same manufacturer were not used because they were not rigid enough. The dowel-rod stems (with leaves attached) were then affixed in a radial fashion to phenolic floral foam blocks (3.81 × 7.62 cm, H × rad) (Flofare, Decorat LTD, unknown city, Mexico) with approximately even spacing in between each stem. The respective numbers of leaves used to form LOW, MED, and HIGH treatment combinations are presented in Table 1.

The floral foam “blocks” (with leaves inserted by their stems) were then stacked to form the respective treatment levels. Three small or three medium-sized leaves inserted into a floral foam block constituted a “half-block”, and six-leaves constituted a “full-block”. The arrangement of blocks on the stack was offset relative to one another to minimize the amount in which leaves overlapped spatially. In addition, blocks of small leaves were always placed above the medium on the stack to mimic “new growth” of the fake plant. The entire stack was secured by plunging a singular dowel rod vertically through the core of the blocks. The arrangements were placed atop inverted nursery pots (15.25 cm D, 1.89 L) (Oubest, Fuzhou City, China), to mimic a potted plant. Controls only had inverted nursery pots.

### 2.3. Leaf Area

An estimate for the total added surface area was calculated with respect to cage volume (0.93 m^3^) by importing photographs and the reference scale from the manufacturer’s e-commerce website (Amazon.com) into ImageJ (Version 1.5h, BSD-2 Public License, NIH). Using the GUI-inbuilt measuring tool, leaf lengths were measured from the tip to the petiole along the midrib. Widths were measured from the widest portions of the margins, perpendicular to the midrib. The top (adaxial) surface area was then approximated as the simple multiple of these two linear dimensions without subtracting any negative space for scalloped edges or lobed leaves. To yield the net total added area, this dimension theoretically could be doubled; but was not in our calculations, since flies almost never utilized the bottom (abaxial) surface of the leaf to perch. Similarly, the Neotropical dung beetle *Canthon septemmaculatus* (Larr.) (Coleoptera: Scarabaeidae) only utilizes the top surface of leaves, albeit for thermoregulation [37].

### 2.4. Rearing

Adult flies were reared starting with 7 d-old larvae acquired from EVO Conversion Systems LLC (College Station, TX, USA) [45] according to the methods of [43], with the following modifications:The first allotment of Gainesville diet [46] (Producers Cooperative Association, Bryan, TX, USA) provided to the larvae weighed 5 kg, and the second weighed 3 kg, rather than an even 4 kg–4 kg split.During the grow-out periods, the walk-in incubator (although set to 26 °C) experienced markedly different conditions (trial 1: 23.2 ± 2.0 °C, 71.7 ± 9.8% RH; trial 2: 25.7 ± 1.5 °C, 53.5 ± 9.7% RH). This information was collected using a HOBO^®^ data logger MX1104 (Onset Computer Co., Bourne, MA, USA). The second trial overlapped with an abnormal winter climactic events (Jan 12–16; Feb 10–11), during which sub-zero temperatures affected most of the continental United States.In addition, after sexing flies according to external genital anatomy [47] and temporarily placing flies into BugDorm-1 holding cages (30 × 30 × 30 cm) (MegaView Science Co., Ltd., Taichung, Taiwan), the thoraxes of females only received a single dot of acrylic paint from a 3 mm-tip Garde’n’ Craft Fine Point Marker (Uchida of America Corp, Torrance, CA, USA) [48,49]. Prior validation experiments show no negative effect against fecundity or fitness of painting females individually in this manner [48]. The handling of flies prior to experimentation also has no reported detrimental effect on fitness [50].

### 2.5. Sequence of Experiment

On the evening prior to the start of each trial, each 0.93 m^3^ cage was stocked with 90 unpainted males (2–4 d-old) and 90 painted females (1–3 d-old). At the top of the hour, from 7:00–18:00 h, behavioral observations (i.e., counts of mating events, count of ovipositing females in traps, counts of males and females occupying perches within each cage) were made by visiting cages in a random order and alternating between two observers for 6 to 7 d. After this duration, the Pareto distribution for mating and oviposition events plateaued, which indicated that little additional behavioral information would be collected. This resulted in a minimum of 72 observations per trial that would then be averaged by day or by hour during statistical analyses. Each cage was misted three times daily with 100 mL of reverse osmosis (RO) water (07:00 h, 12:00 h, and 18:00 h).

One the day following peak mating (i.e., total matings across all cages combined had declined), which indicated the initiation of the post-mating interval, an attractant box and egg trap were each added to each cage [41]. This set-up consisted of plastic shoeboxes measuring 35.6 (L) × 20.3 (W) × 12.4 (H) cm (Sterilite Corp., Townsend, MA, USA) containing 1 kg of a 70% moisture Gainesville diet inoculated with an aliquot of ~1000 7 d-old larvae. Lids were modified to allow volatiles to escape through a 12.7 (L) × 5 (W) cm rectangular slot, which was replaced with nylon screening. Egg traps were constructed by taping together three strips of corrugated cardboard, measuring 10 (L) × 3.5 (W) × 1.25 (H) cm, and placed atop the lids. Previous work indicated that delaying substrate provisioning can have a positive effect on trapped eggs in conjunction with a restricted cohort age [43].

Cardboard egg traps were replaced daily at 12:00 h [41]. Eggs were then harvested following the methods of [43]. In addition to the total weight of eggs per trap per cage per day, the number of clutches in each trap was also recorded to potentially better indicate the number of oviposition events that had occurred, as previous work also tended to show a lack of correlation between oviposition counts and egg weights [43]. Any collected eggs were then placed in 30 mL solo cups and glass canning jars following the procedure from [41,42] and incubated within the same walk-in chamber (trial 1: 27.2 ± 1.2 °C, 50.0 ± 7.3% RH; trial 2: 25.6 ± 1.2 °C; 51.2 ± 5.6% RH). All collected eggs were monitored thereafter for five days [43] for the hatch rate [41,42].

### 2.6. Non-Parametric Statistics

All statistical analyses were performed in RStudio (version 4.1.2) using tidyverse packages [51]. The visual inspection of data followed by the Shapiro–Wilk test for normality confirmed that the distributions of data (mating events, oviposition events, number of clutches, clutch weight, egg weight, and percent hatch) were non-normal (in all cases being highly skewed or zero-inflated). Therefore, nonparametric analyses (i.e., Kruskal–Wallis *H*) were used to test whether treatment or increasing the leaf area influenced fitness metrics, with an a priori alpha criterion of 0.05.

### 2.7. Perch Disparity Metric

Because painting females revealed obvious sex-specific behavioral patterns during experimentation, it was necessary to develop an index to describe perching patterns with respect to sex, which we call perch disparity.

Here defined, perch disparity is the simple difference in the count of females perching (per cage/per day) minus the count of males perching (per cage/per day), where values greater than 1 indicate a female bias, 0 indicates parity, and negative values indicate a male bias. Values for perch disparity were automatically calculated in Excel (version 2403, Build 17425.20176) by subtracting column totals from one another.

Untransformed perching data for each trial were cleaned, plotted, and fitted to a curve using MS Excel’s built-in graphing tools. Regressions were selected based on the equation that maximized the R2-value while using the least number of terms. If this could not be achieved, then the simplest expression was chosen. When calculating the area between two curves, we verified the difference of the two integrals using Wolfram Alpha (Wolfram Alpha LLC, Champaign, IL, USA).

### 2.8. Time Series

To conduct a time-series analysis, missing perch disparity data from unobserved time periods were imputed into MS Excel by calculating a 13 h moving average. The imputed values were then superimposed onto the original function to create a continuous “observed” sequence.

The data sequence was then coded as a time series and a time-series analysis was conducted in R (version 4.1.2) using the TTR package and the decompose() function. This function separated the waveform into its constituent components: trend, seasonality, and random noise. Trend represented week-long patterns in behavior, seasonality was daily patterns, and random noise was the remaining discrepancy between these two combined effects and the observed sequence.

## 3. Results

### 3.1. Kruskal–Wallis H

The distributions of fitness metrics generally followed non-Gaussian distributions (see Appendix A), and so this required analysis using non-parametric methods. Treatment had no statistical effect on any tested fitness metrics (Kruskal–Wallis *H* Test) (Table 2). Therefore, conducting Dunn’s test with Bonferroni Correction as a post-hoc test was unnecessary.

### 3.2. Predictive Model and Extrapolation

Because there was no effect of treatment, either negative or positive, it was reasoned that increasing the surface area available for flies could be a method to increase the number of flies relative to the cage volume in a strict sense. Therefore, total perching per cage was summed and fit to a linear model with a Gaussian distribution to the data to both interpolate and extrapolate how many flies might be added to cages while maintaining the same level of perching behavior (Figure 1).

The equation for the model can be given as follows:Expected Perching Flies = 6.74 + 38.84 × Leaf Area (m^2^ per 0.93 m^3^) − 0.08 ∗ Day(1)

Model parameters (Table 3) were selected by first increasing the *R*^2^ value (to 0.78) of a linear model and then minimizing Akaike’s Information Criterion (AIC) for the equivalent general linear model (Gaussian family with identity link). This outcome was achieved by (a) excluding totals from trial 2, (b) including perching totals from both sexes, and (c) including day as a covariate. Non-significant terms were excluded to keep the model parsimonious. The number of flies was also normalized (i.e., 100 flies of each sex per 0.93 m^3^ cage) to align with previous studies [40,41,42]. Results were interpolated up to a surface area of 1.00 m^2^ (Table 4).

### 3.3. Perch Disparity

While sex disparity was in a strict sense female-biased (Table 5), this phenomenon was not static and instead changed over time. On average, the difference between the mean number of females perching on plants and mean number of males perching on plants (defined as “perch disparity”, see Methods) decreased in an absolute sense both over successive days (Figure 2) and hours (Figure 3), towards parity, becoming more male-biased. For each day of trial 1, the mean (n = 3) perch disparity became more male-biased by 5.78. From 0700 h to 1800 h, disparity also shifted towards an additional 2.29 males per cage, per hour on average (n = 3). In trial 2, sex disparity was initially female-biased but quickly shifted towards male bias starting on day 2 (Table 5), likely due to high female morbidity (see Section 4).

Because counts of perching flies were predicated based on flies being there to be observed, results were likely impacted by density effects (e.g., morbidity/mortality), which were in turn affected by environmental conditions. Specifically, during trial 2, 63.6 females died per cage by the end of the experiment on average, whereas just 26.6 males died per cage on average over the same period, representing a 2.39-fold difference (Appendix A). Although there was an obvious trial effect, which caused perch disparity to be much more male-biased in trial 2 than in trial 1, both treatment and temporal patterns were qualitatively consistent between trials (Figure 4). Hence, to approximate a scaling factor between trial 1 and trial 2, a logarithmic regression was fit to both sets of data and the intercepts were compared.

Subtracting the two intercepts yielded indicated that an equivalent cage would have been expected to have 57.9 more perching females than males per day in trial 1, as compared to trial 2 (Figure 4). Time-series analysis was then conducted on trial 1 data only (Figure 5) to be more robust to random effects (see Appendix A). From this, results could be generalized from trial 1 to trial 2 by incorporating the scaling factor of 57.9.

### 3.4. Time Series

Time-series analysis (Figure 5) showed that this decline is only temporary and that female bias recovers by the beginning of the observation period each day. The weekly trend had relatively little change in perch disparity during the first three observed days, followed by a large dip towards parity on the third day, a large resurgence in female bias by day four, and then a quick decline in perch disparity that finally reached parity for the remaining days of the experiment.

## 4. Discussion

By introducing artificial plants of different sizes into breeding cages, this experiment investigated how modifying the available surface-area-to-volume ratio can affect fertile egg production of the adult black soldier fly. Because females from males were differentiated by marking their thoraxes with acrylic marker paint [48], sex-specific perching behaviors were documented for the first time in captivity. We hypothesized that a medium amount of additional surface area relative to the cage volume would increase fertile egg production but that too much would have a negative effect. In addition, we also thought that males would primarily be those interested in defending territories immediately around perches [29]. However, treatment had no statistical effect on any recorded fitness metric, either positively or negatively. This suggests that by increasing the surface area, additional flies can also be added [31] without a negative effect on improving egg production per cage. Hence, a linear model was developed that could extrapolate the potential additional number of flies that could be accommodated by increasing the surface area within cages.

Surprisingly, females engaged in perching much more often than males, though this pattern was dynamic and shifted towards parity throughout time (week/day). In addition, a trial-effect revealed that, in a strict sense, perch disparity (the difference between the number of females and males perching on plants) was density-dependent and strongly affected by environmental conditions. Such conditions were thought to likely have increased morbidity disproportionately in females, which was observed in trial 2. Discounting such effects, remaining differences in perching behavior between the sexes (viz. those observed in trial 1) are likely due to underlying sexually dimorphic neurophysiology [51] and functional anatomy [43]. Specifically, males are known to have higher counts of brain cells within their optic lobes [51], which may correspond with mate location/recognition, such as through sexually dimorphic body size [40] or wing-interference patterns [52]. In addition, there are sexual dimorphisms present in the antennae, with males having a larger pedicel [53], which houses the Johnston’s organ. This organ is responsible for the detection of airspeed and coordination of in-flight motion [43]. By contrast, female antennae have longer flagella [53], a region that features dense concentrations of sensory cells (flagellum), which would potentially correspond to an enhanced ability to detect volatiles at oviposition sites [53] emanating from the substrate and/or conspecifics [54,55].

This report is especially relevant for producers that are interested in promoting natural behaviors via an insect welfare-minded approach [32,56] because presently, it is generally unknown which individuals engage in mating versus resting behaviors, at which times during the reproductive cycle, or how these behaviors might be influenced by the physical structure of the artificial rearing environment.

### 4.1. Spatial Complexity

Despite our initial hypotheses, this experiment does not provide strong evidence for a link between habitat structural complexity and fitness [39]. We concluded this because mating events, oviposition events, the number of clutches, the mean clutch weight, the harvested egg weight, and the hatch percentage (all measured per cage/per day) did not differ significantly across treatments. This finding was contrary to what has been observed in some other flies, particularly mosquitoes, like *Aedes agypti* (Linnaeus *in* Hasselquist, 1762) [57] and *Anopheles gambiae* Giles 1902 (Diptera: Culicidae) [57]. For these species, males form lek-like aerial swarms [26] near specific physical structures called swarm markers [57], such that the physical environment can influence variation in mating success. We discuss several potential reasons for this.

First, treatments were designed to crudely approximate the fractal geometry of biological structures and ecological habitats [58], rather than test the multi-dimensionality of the true spatial complexity. Fractal geometry can be seen in the organization of plant circulatory systems [59], since limbs and roots can continually be sub-divided into self-similar parts irrespective of scale. However, this is only one dimension of spatial complexity, which can technically vary based on three separate axes [58]: (a) heterogeneity, or variation per relative abundance, (b) complexity, or variation per absolute abundance; and (c) scale resolution [58]. Hence, some aspects of habitat spatial complexity were not represented in our design. Specifically, our set-up featured plant leaves that were identical copies of one another, and each treatment level was merely a doubling of scale from the previous without a change in the spatial conformation. Research on the perching of Neotropical dung beetles (Coleoptera: Scarabaeidae) suggests that leaf preference varies by insect species and is correlated with individual size, with larger individuals perching on higher and larger leaves [38]. Because no functional response in terms of fitness was found to simply increase the amount of available surface area, this suggests that black soldier flies may instead be more responsive to either an untested structure or increased range of structures possibly owing to their Neotropical origin [44], where plant structures are hyper-diverse [60].

Second, the level of the additional surface area relative to the cages (84 × 84 × 133 cm, L × W × H) may simply not have been enough to produce a response to begin with, either positively or negatively. This is because the added surface area for all treatments was less than five percent of the total surface area of an empty cage despite occupying a substantial volume. This is supported by an industrial-scale study that tested an additional surface area of 3.33, 3.99, 5.11, and 7.17 m^2^ per 1.00 m^3^ cages at a density of 16,000 flies [61]. The researchers found that the 3.99 m^2^/m^3^ treatment lead to an increased yield of 18.61 ± 4.42 g of eggs compared to 9.23 ± 3.14, 15.00 ± 3.10, and 11.30 ± 3.62 g of eggs for the 3.33, 5.11, and 7.17 m^2^/m^3^ treatments, respectively [61]. This meant that, for their system, an increase in 0.6 m^2^/m^3^ corresponded to a 2.01-fold increase in egg production.

Hilltopping butterflies (Lepidoptera) reportedly avoid areas that are hyper-dense with males and lay few eggs in areas that are spatially bare [35]. Thus, increasing spatial complexity beyond the range that was tested in this study could promote reproduction by reducing perceived competition and the harassment of mated females [27,35]. While it was hypothesized that too much additional surface would limit reproduction, the potential for artificial plants to be obstacles (visually or spatially) at the scale tested was likely too low.

As fliers, black soldier flies can reposition themselves easily. The amount of additional surface area needed to create a visual or physical barrier when provided by the topology of artificial plants (as opposed to a single, continuous surface—e.g., a wall) is likely much larger for flying insects than for insects that locomote on the ground. Consider that for other farmed insects, like crickets (Orthoptera: Ensifera), e.g., *Ruspolia differens* (Serville, 1838) (Orthoptera: Tettigoniidae) [62], an egg board is added to reduce crowding and provide a visual obstruction to prevent cannibalism [62], but these insects primarily locomote by crawling or jumping over short distances and so are not as vagile as black soldier flies.

Third, the choice to use artificial plants over real plants may have removed several elements from the system that are potentially critical for connecting spatial complexity with fitness. These include green-leaf [63] and floral [64] volatiles (which are widely known to attract and influence insect behavior), the presence of bacteria (e.g., which the sexually mature male and immature females of flies of the genus *Dacus* (Diptera: Tephritidae) are known to consume [65]), and different color and thermal properties that can often be non-apparent to humans. While few reports describe plant–insect interactions between adult black soldier flies (e.g., defending territories around Kudzu, *Pueraria montana* (Loer) (Merr.) [29], the numerous instances of community science members posting images of black soldier flies on (unidentified) plants (e.g., iNaturalist, BugGuide, LinkedIn, etc.) point to the obviousness and plethora of the interactions that must exist in nature but at present stand to be documented. Within captivity, the effect of materials (with different colors) used as artificial perches on total egg production has been investigated [61], with results showing increased performance when using a white material (19.4 ± 4.9 g eggs/cage) compared to green (15.0 ± 2.1 g eggs/cage) or a mix of the two (10.4 ± 1.5 g eggs/cage), although there was no significant difference between white and green.

Critically, while the use of real plants theoretically could buffer against environmental stressors by providing shade and reducing critical temperature thresholds—their use, unfortunately, adds an additional layer of complexity and perhaps an unwarranted risk [66], since one of the key issues of growing plants indoors is the threat of pests and pathogens invading without natural enemies to combat them in a relict environment. The choice to use artificial plants was necessarily born out of the desire to make the experiment replicable and the habitat easier to clean and mirrors the present Zeitgeist that the transmission of zoonotic pathogens will, for the foreseeable future, continue to be an issue that affects many organisms living in confined facilities (e.g., confined animal facility operations [67]).

### 4.2. Predictive Model

Because no negative effect of including additional surface area was determined, a linear model was fitted to the data to relate the number of perching flies to increases in the leaf area. The extrapolation of model results yielded that, for example, increasing the surface area by 1.00 m^2^/0.93 m^3^ is expected to have the effect of providing enough additional perching area for 45.6 flies at a 1:1 sex ratio. While the model was normalized for a density of 200 flies (1:1 sex ratio), it was not normalized to a volume of 1.00 m^3^. This was because the observed patterns, or lack thereof, were likely specific to the cage dimensions tested. Adding the equivalent surface area to a cage with a larger volume or different dimensions is likely to yield different results due to sex-specific lekking behaviors [5] and the effects of scale. For instance, the 0.93 m^3^ cage used in this experiment is an order of magnitude larger than, for example, the smallest 0.02 m^3^ cages used by some researchers [68]. It is comparable to some cages used by some industry, while being a third of the size of the relatively standard 2.88 m^3^ cages (EVO Conversion Systems, LLC, Bryan, TX, USA) and two orders smaller than some of the larger 59.42 m^3^ cages available to industry (InsectoCycle, Gelderland, ND, USA). The model results additionally suggest that as days progress, the number of perching flies (viz. females) is expected to diminish slightly, which can be explained by flies expending energy and becoming less active before dying (personal observation). The precise rate and shape of decline predicted by the model is likely specific to the environmental conditions (see next section) and local adaptations of the tested adult population, but qualitatively should be expected to be generally similar across black soldier flies. For this reason, the results of the model need to be validated independently across a variety of scales.

Additionally, because artificial plants were placed on the bottom of cages and leaves were arranged to maximize the exposed surface area, the results should also be interpreted conservatively because it can be reasonably assumed that flies would have otherwise been able to rest on the floor of the cage. For this reason, future designs should be mindful of ways that the additional surface area effectively replaces a portion of the existing surface area (rather than expanding it) so that flies have the most opportunity to utilize any provided perches. Simulations should also be developed to investigate economic trade-offs between upfront material costs and ongoing labor costs, against any increase in egg production.

Future behavioral research is poised to explore ways to biologically sort flies through the interaction of behavior with the microclimate [69], for instance, if it is shown that flies vertically stratify based on age or condition within cages, with perhaps the largest, youngest, or most-fit flies preferring different perching locations or temperature-humidity ranges. Similar patterns are well-known/documented in a variety of animals, including open-cage and free-range chickens, *Gallus domesticus* (Aves: Galliformes) [70], where birds self-stratify based on dominance hierarchies [71]. Because insect interactions are size-structured [40,72,73], there is reason to suspect that the same can occur for the adult black soldier fly. For this reason, future experiments should consider alternative hypotheses in which flies use all cage surfaces equally or segregate into preferred areas based on the underlying biology.

### 4.3. Perching and Perch Disparity

When considering raw counts alone, females are the sex that primarily occupied perches. This finding seems to contradict a previous description [29] and the general belief held by black soldier fly researchers (personal communication) that males are those that occupy and defend perches to maintain leks. Certainly, this response may have been an artifact of the mating cage being restrictive compared to the amount of space black soldier flies likely utilizes in the wild [5]. Previous studies on perching in insects have documented that males utilize perches to investigate similar-sized objects (i.e., conspecifics) to themselves [35]. When males encounter other males, this results in brief conflict followed by separation [35]. When males encounter females this generally results in mating [35]. Similar patterns have been observed in black soldier flies [29]. Casual observations during our experiment revealed that female black soldier flies intermittently join aerial swarms before returning to their perches.

Returning to the same spot has been described as an energy-saving mechanism because flying to a new spot may result in unwanted harassment [35]. More generally, insect flight is a highly energetically taxing activity [74] that trades off with egg production [75,76,77]—although different fats are generally used for each (triglyceride vs. phospholipid) [78]. Hence, reason dictates that perhaps it is females that gain the most from an increase in the available surface area. Sit-and-wait strategies, like perching, theoretically allow females to reduce total energy usage [79], especially when likely a great deal is needed to constantly maintain flight to and from lekking sites [5,80]. We speculate that the ability to temporarily rest on plants might offer females the ability to preserve precious energy stores while gauging the suitability of mates [35], whereas for males, constant jostling for position amongst rivals may not have selected for this behavior [81]. Therefore, perching may promote lekking behavior by allowing females a greater opportunity for choice within leks [24], though this needs to be verified. Such behavior can be contrasted with that of *Aedes* mosquitoes, which, although engaging in aerial swarms, have limited opportunities for precopulatory female choice other than tarsal kicks (i.e., “bucking”) [82].

When considering patterns over time in sex disparity data, perching behavior aligns with previous conceptualizations. That is, mating intensity generally peaks in early hours (e.g., by 12:00 h under artificial light [83] or 15:00 h in natural light [84]) and also by the third day of experimentation [40]. The amount of mating also negatively regressed with the time of day [84]. Although sex disparity is clearly female-biased, the strength of this bias is dependent on time and wanes over the course of the day, suggesting the following: (a) females could be leaving perches to join mating swarms, or (b) males could be leaving mating swarms to join perches. Both are potentially supported because there are several cases when total perching by either sex was increased relative to the preceding day or hour (Table 5 and Table 6).

Critically, the presence of a trial effect revealed that an instantaneous operational sex ratio (i.e., the number of males versus females at minute points in time) influenced behavioral patterns, which is in turn affected by the morbidity effects caused by suboptimal abiotic conditions [35]. In our experiment, although all populations started at an equal 1:1 sex ratio, trial 2 had a much higher female mortality rate by the end of the experiment (Appendix A). Each trial experienced differential abiotic effects after the initiation of behavioral observations that likely then caused bottom-up effects on the population structure and behavior, potentially because climactic conditions external to the indoor rearing environment caused the central heating systems to activate more frequently, lowering the relative humidity (see Appendix A).

Specifically, during the grow-out phase, there was a large difference in the temperature and humidity that flies experienced across trials (i.e., 22.0 °C median temp and 72.3% median RH in trial 1, compared to 26.2 °C median temp and 54.1% median RH in trial 2). Because of this, although the flies used in the experiment were, in a strict sense, the same temporal age, they were unlikely to be the same biological age since both temperature [7,72,85] and humidity [8,9,86] affect the rate at which insects develop. This means that the flies in trial 2 were likely in effect much older relative to those used in trial 1 (based on relative humidity days) since they experienced more heat and drying days. In addition, after being introduced to cages, although starting at the same RH across trials, the breeding environment in trial 1 was increased in relative humidity, while the breeding environment in trial 2 was decreased in humidity, compounding the difference by 14.7 RH-days (Appendix A).

Past studies have shown differing results regarding longevity with either (a) females living ~3.5 less days than males, irrespective of the rearing temperature, or (b) no difference between males and females [87], potentially due to the ability of poikilotherms to thermoregulate [72]. Alternatively, another study suggested that, by mating, females increase their longevity relative to males by gaining nutrition via nuptial gifts [88]; this is potentially supported by our experiment, especially if the extension of female longevity is predicated on females mating early in their (biological) lives. If flies were held in holding cages and were delayed in mating [41], perhaps this then explains the increased mortality of females in trial 2; however, no such effect has been seen in other experiments on black soldier fly aging [41,43].

This study reveals that behavior is complex and not easily intuited. There is still much to be understood, even in environments that are purportedly environmentally controlled [89]. As the field continues to progress and describe these multi-tiered interactions, it is important that both theory and empiricism continue to be linked and inform one another.

### 4.4. Lekking?

While a 0.93 m^3^ cage has often been thought to be too crowded to promote lekking behavior [5], the provisioning of our experimental cages with artificial plants provided the opportunity to show that perhaps at least some elements of black-soldier-fly-lekking behavior is preserved within captivity, especially in larger cages; though this does not rule out the possibility that lekking is being selected against in favor of something akin to scramble-competition polygyny [81,90].

Identifying criteria [23] for lekking have been formalized and applied to insects [24,26], including the black soldier fly [28,29], the closely related *Hermetia comstockii* Williston (Diptera: Stratiomyidae) [73], and orchid bees (Hymenoptera: Apinae: Euglossini) [91]. The aspects of lekking that seem to be preserved for the black soldier fly in captivity include the spatiotemporal segregation of males and females, since males primarily occupied aerial swarms during the early hours and females were those that primarily occupied perches. In addition, males engaged in prolonged bouts of mating behavior near lights, as indicated by casual observations. These mating swarms occurred in cages regardless of whether perches or attractant boxes were present, suggesting that the mating system is unlikely to be akin to resource-defense [92] or female-defense polygyny. Instead, the structure of black soldier fly leks (i.e., their dimensions) appear quite plastic and may contract or grow depending on the available space (i.e., exploded vs. imploded leks) [25].

This study’s results also suggest that any positive welfare effect of increasing the surface area will likely be primarily towards females, as when environmental conditions were optimal, they were those that utilized perching structures the most. Future research should aim to devise ethograms (e.g., [27,91,93]) to properly quantify the positive contribution of perches to improving quality-of-life. Although there was not a direct link between fitness and the increased surface area quantified (while despitekeeping the population density constant), the placement of the artificial structures allowed for the re-conceptualization of what can occur within the confines of even a 0.93 m^3^ cage, such that the lek-mating system may still persists within captivity for black soldier flies (for now) despite the pressures of artificial selection [18]. Indeed, lekking may be encouraged through the modification of the artificial habitat [22]. It is possible that with a larger space, separate areas can be devised for each sex—one in which males can engage in mating bouts, and another for females to rest, continue their reproductive development, and lay eggs. Certainly, prior work has implicated the presence of old males in swarms to either have no positive or a net negative contribution to fitness [41,43], and so, devising ways to isolate the non-reproductive from the reproductive flies within cages will likely promote the applied aims of obtaining consistent egg production.

## Figures and Tables

**Figure 1 insects-15-00770-f001:**
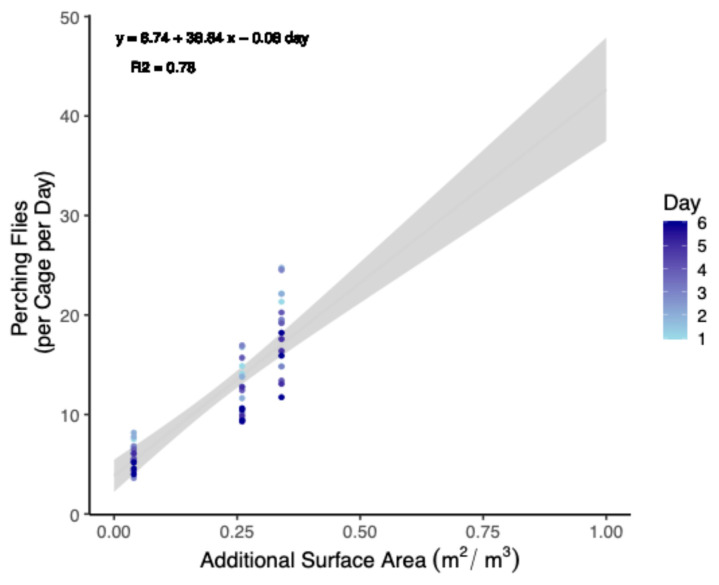
Predictive model relating increases in leaf surface area to the percent increase in the number of perching black soldier flies (Trial 1). Artificial plants were provided to 0.93 m^3^ cages up to a maximum value of a 0.33 m^2^ adaxial surface area. Beyond this, values are extrapolated. Data were also standardized from populations of 90 males and 90 females to 100 of each. The grey area indicates a ±95% confidence interval.

**Figure 2 insects-15-00770-f002:**
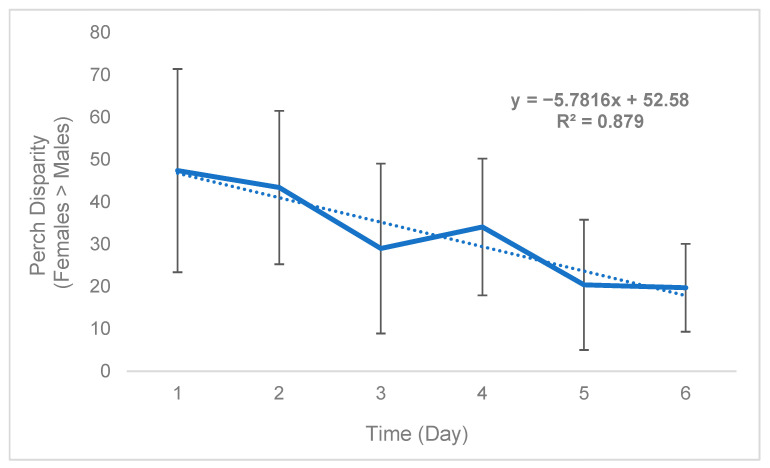
Black soldier fly mean daily perch disparity over time (Trial 1). Perch disparity was calculated as the sum of females perching minus the sum of males perching. Error bars indicate ±SD. The best-fit line is a linear regression. The experimental unit was a 0.93 m^3^ mating cage (n = 4 treatments, n = 3 replicates) held within an indoor rearing environment in Texas, USA. Each cage had an initial population of 90 males and 90 females.

**Figure 3 insects-15-00770-f003:**
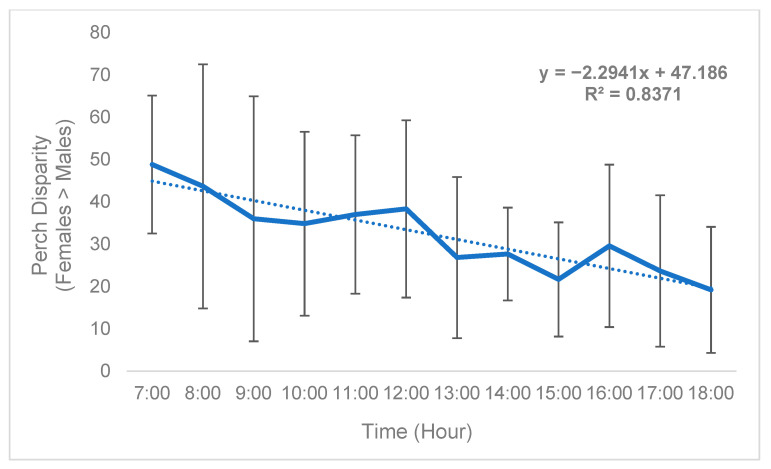
Black soldier fly mean hourly perch disparity over time (Trial 1). Perch disparity was calculated as the sum of females perching minus the sum of males perching. Error bars indicate ±SD. The best-fit line is a linear regression. The experimental unit was a 0.93 m^3^ mating cage (n = 4 treatments, n = 3 replicates) held within an indoor rearing environment in Texas, USA. Each cage had an initial population of 90 males and 90 females.

**Figure 4 insects-15-00770-f004:**
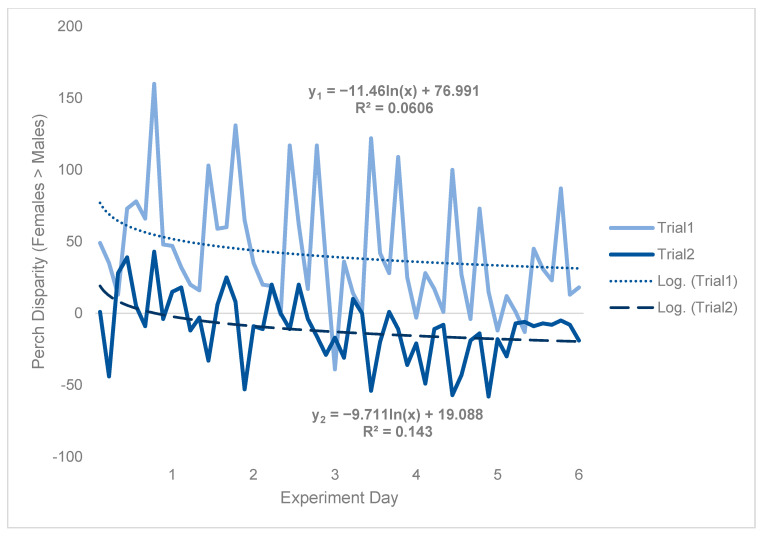
Black soldier fly perch disparity with respect to each trial. Perch disparity was calculated as the sum of females perching minus the sum of males perching, with an x-intercept representing parity. Data were collected in 0.93 m^3^ mating cages (n = 4 treatments, n = 3 replicates) held in an indoor rearing environment from 07:00 h to 18:00 h for 6 days. Each cage had an initial population of 90 males and 90 females. Gaps in observation were not filled in using a moving average. The best fit lines are logarithmic regressions, with y_1_ indicating trial 1 and y_2_ indicating trial 2.

**Figure 5 insects-15-00770-f005:**
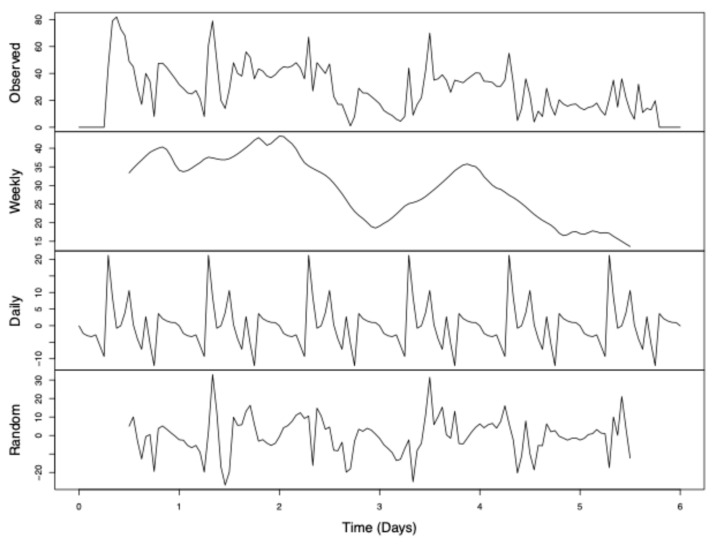
Deconstructed time-series analysis of perch disparity. Perch disparity was calculated as the sum of females perching minus the sum of males perching. Data were collected in 0.93 m^3^ mating cages (n = 4 treatments, n = 3 replicates) held in an indoor rearing environment from 07:00 h to 18:00 h for 6 days. Each cage had an initial population of 90 males and 90 females. Gaps in observed data were filled by taking a 13-hour moving average. Observed data are the sum of “trend” + “seasonal” + “random” data. Respectively, these represent the following: (a) declining perch disparity with each day and a secondary post-mating peak; (b) cyclical pattern of sex bias to peak in the early hours and decline throughout the day; and (c) random effects not otherwise explained.

**Table 1 insects-15-00770-t001:** Description of treatment levels.

Level	Construction	Estimated Adaxial Leaf Area (m^2^)
CTRL	Inverted pot only	0.000000
LOW	Pot + ½ small block + ½ medium block	0.040061
MED	Pot + 1 small block + 1 medium block	0.260011
HIGH	Pot + 2 small blocks + 2 medium blocks	0.340133

**Table 2 insects-15-00770-t002:** Kruskal–Wallis *H* Test of fitness vs. treatment level (leaf area).

Metric	Chi-Squared	Degrees of Freedom	*p*-Value	Significance
Mating Events	0.51983	3	0.9145	n.s.
Oviposition Events	1.96350	3	0.5800	n.s.
Clutches	0.41339	3	0.9375	n.s.
Clutch Weight	0.13509	3	0.9873	n.s.
Egg Weight	0.37939	3	0.9445	n.s.
Percent Hatch	0.62169	3	0.8914	n.s.

“n.s.” indicates *p* > 0.05.

**Table 3 insects-15-00770-t003:** Summary table of linear model.

	Estimate	Std. Error	T Value	Pr(|t|)	Significance
Intercept	6.7416	1.0416	6.472	3.71 × 10^−8^	***
Leaf Area	38.8370	2.8975	13.404	<2 × 10^−16^	***
Day	−0.8280	0.2152	−3.845	0.000334	***

*** indicates Pr(|t|) < 0.001.

**Table 4 insects-15-00770-t004:** Additional flies expected to be accommodated with additional surface area provided within a single 0.93 m^3^ breeding cage. Values were generated from a predictive model that related increases in added surface area to perching observation counts, with respect to each day of the experiment. Data was standardized to initial population of 100 flies of each sex.

Added Surface Area (m^2^/m^3^)	Day 0	Day 1	Day 2	Day 3	Day 4	Day 5	Day 6
0.25	16.45	15.62	14.79	13.97	13.14	12.31	11.48
0.50	26.16	25.33	24.50	23.68	22.85	22.02	21.19
0.75	35.87	35.04	34.21	33.39	32.56	31.73	30.90
1.00	45.58	44.75	43.92	43.09	42.27	41.44	40.61

**Table 5 insects-15-00770-t005:** Sum of observed female and male perching events per day across n = 3 replicate cages, for each trial. * Indicates in increase in total perching compared to the previous day.

	Trial 1		Trial 2	
	Female	Male	Female	Male
Day 1	935	366	366	276
Day 2	961 *	440 *	285	338 *
Day 3	779	431	176	224
Day 4	771	396	184 *	346
Day 5	634	389	88	365 *
Day 6	595	378	25	124
Total	4675	2400	1124	1673

**Table 6 insects-15-00770-t006:** Sum of observed male and female perching events per hour across n = 3 replicate cages, for each trial. * Indicates an increase in total perching compared to the previous hour.

Time (Hour)	Trial 1		Trial 2	
	Female	Male	Female	Male
7:00	501	208	128	145
8:00	500	238 *	134 *	128
9:00	490	274 *	120	127
10:00	458	249	110	127
11:00	509 *	287 *	124 *	163
12:00	490	260	103	139
13:00	372	211	70	164 *
14:00	358	192	76 *	196 *
15:00	289	159	84 *	203 *
16:00	249	101	70	134
17:00	265	123 *	67	125
18:00	194	98	128 *	145 *

## Data Availability

The original contributions presented in the study are included in the article/Appendix A; further inquiries can be directed to the corresponding author.

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
