# Peer review of "Sex-Specific Perching: Monitoring of Artificial Plants Reveals Dynamic Female-Biased Perching Behavior in the Black Soldier Fly, Hermetia illucens (Diptera: Stratiomyidae)"

_insects, 2024, doi:10.3390/insects15100770_

Round 1
Reviewer 1 Report
Comments and Suggestions for Authors
Reviewer Report: Sex-Specific Perching: Monitoring of Artificial Plants Reveals Dynamic Female-Biased Perching Behavior in the Black Soldier Fly, Hermetia illucens (Diptera: Stratiomyidae)
Overall Recommendation:Publish with minor revisions.
Strengths
Novelty:The study investigates a previously unexplored aspect of black soldier fly (BSF) behavior and reveals a clear sex bias
.
Clear Methodology: The experimental design is well described, including cage size, fly densities, perching area variations, and data collection methods
.
The results demonstrate a significant and dynamic difference in perching behavior between male and female BSFs. The daily and age-related patterns are particularly interesting
.
Practical Applications, The findings have potential applications for BSF mass-rearing practices by informing cage design and optimizing resource allocation.
Weaknesses:**
Limited Explanation for Sex Bias:
minor typing in the manuscript
Suggestions for Revision
Expand on the discussion of potential reasons for the observed sex bias in perching behavior. Is it purely related to male swarming, or could there be other physiological or behavioral factors at play
?
**Additional Comments:**
* The simple summary is clear and concise, effectively capturing the key finding for a broader audience.
* The keyword list is comprehensive and relevant to the study.
**Overall, this is a well-conducted study that provides valuable insights into BSF behavior. With minor revisions to address the weaknesses mentioned above, the manuscript would be even stronger and suitable for publication.**
Author Response
Dear Reviewer,
Thank you very much for your review/comments. Adding a section on functional anatomy is a great suggestion/improvement.
Sincerely,
N. B. Lemke
Expand on the discussion of potential reasons for the observed sex bias in perching behavior. Is it purely related to male swarming, or could there be other physiological or behavioral factors at play.
Response: The following has been added:
"... differences in perching behavior between the sexes (viz. those observed in Trial 1) are likely due to underlying sexually dimorphic neurophysiology [51] and functional anatomy [42]. Specifically, males are known to have higher counts of brain cells within their optic lobes [51] that may correspond with mate location/recognition, such as through sexually dimorphic body sizes [39] or wing-interference patterns [52]. In addition, there are sexual dimorphisms present in the antennae, with males having a larger pedicel [53], which houses the Johnston’s organ. This organ is responsible for the detection of airspeed and coordination of in-flight motion [42]. By contrast, female antennae have longer flagella [53], a region which features dense concentrations of sensory cells (flagellum), that would potentially correspond to an enhanced ability to detect volatiles at oviposition sites [53] emanating from the substrate and/or conspecifics [54,55]."
Reviewer 2 Report
Comments and Suggestions for Authors
Review of „Sex-specific perching: Monitoring of artificial plants reveals dynamic female-biased perching behavior in the black soldier fly, Hermetia illucens (Diptera: Stratiomyidae)”
General comments:
This manuscript by Lemke and colleagues focuses on the effect of artificial plants on the sex-specific perching behavior of black soldier fly adults. It is very well written and the hypotheses and conclusions are valid and in line with the results of the study. In addition, the authors identify possible weaknesses in the study design and provide suggestions as to how these may have affected the results.
Please find my specific comments below.
Specific comments:
1) Line 22. …day 1…day 6…
2) Line 34. …biotic and abiotic factors
3) Line 36. detritivores or detrivores
4) Lines 43-45. Also consider adding Nayak et al 2024 (https://www.mdpi.com/2077-0472/14/1/8), where such parameters are mentioned and discussed in detail.
5) Line 63. Please use the “ – “ symbol instead of “-“ for ranges throughout the manuscript.
6) Line 105. …ran on a 12:12 L:D photoperiod…
7) Lines 107-109. Why didn’t you control the temperature and humidity? Please provide the range of both in the M&M section.
8) Line 112. Add space between number and unit
9) Line 113. Why did you use Monstera leaves instead of smaller ones? Please clarify.
10) Line 114 and 119. Add the city of the companies.
11) Table 1. Use an appropriate table heading. Moreover, m2 has to be superscript.
12) Lines 151-156. Use spaces between number and unit (but not for % since no unit). I think one decimal is precise enough, please change accordingly. Since you used an “wal-in incubator”, why are the temperatures and humidities so different between the trials? Is this incubator located outside?
13) Line 175. Add space before “h”.
14) How did you know that the mating peak was reached during the trial? Please clarify.
15) Lines 180-182. Please provide length information in the same way like before. Use the same number of decimals here.
16) Lines 186-189. Did you also consider multiple oviposition events per female?
17) Lines 191-193. Use the same font everywhere. One decimal is appropriate.
18) Were the flies within the cages the same age (synchronized)? How old were they (days after emergence) at the beginning of the experiments? Did you exclude individuals that died within the first 1-2 days after the start of the experiment?
19) Line 190. Do you mean ml?
20) Line 217-218. How large do you estimate the error that is introduced into the data by the unobserved time periods?
21) Table 2. Please add that df means degrees of freedom. Use the same number of decimals for the respective metric described.
22) Line 246. Mention that CI means confidence interval.
23) Line 248. Use superscript for m3.
24) Table 3. You mention once ** and once *** for the same value. Please revise accordingly. Please provide a more detailed table caption.
25) Table 4. Change formatting of the left heading cell. Add a space between unit and number. Use the same number of decimals within the table.
26) Line 259. Which supplementary exactly?
27) Lines 266-267. …towards male bias on day 2 and what?
28) Table 5. Please highlight Trial 1 and 2 (better overview) and delete unnecessary lines in all tables.
29) Figures 2 & 3. Please add axis lines and auxiliary lines to the scaling. Add axis labeling in Figure 3.
30) Please make sure that there are spaces before and after a “=” symbol in the entire manuscript (figure captions, text, tables,…).
31) Figure 5. Note the capitalization of the axis label (Observed, Trend, Seasonal, Random). Please add the unit “days” to Time. Moreover, you made a mistake while numbering the figures (figure 5 before figure 4), please change accordingly and check the text.
32) Figure 4. Please change according to my comment 29). In the figure caption, please add “fly” after Black soldier. Why did you not filled in gaps here?
33) Lines 303-307. One decimal is appropriate. Sometimes you write the trials capitalized, sometimes not. Please make sure to keep it the same throughout the whole manuscript. Also add exact supplementary file(s) in line 306.
34) Line 309. Which declination? Since you started a new chapter, you should provide more information what you are writing about.
35) Lines 325-326. What would be the consequence for industrially insect farming?
36) Line 346. Provide order and family.
37) Line 367. Add space after reference.
38) Line 369. Delete the first “surface”.
39) Line 374-378. Use superscript numbers.
40) Line 385. Delete one of the “easily”.
41) Did you found differences in longevity of the adults? Could this factor play a role?
42) Line 394-405. What is actually known about (specific) volatiles that can be detected by the BSF adults? And what is known about color preferences?
43) Adding surface area (e.g. by artificial leaves) could therefore enable a higher density of flies/cage, which is promising. How do you evaluate the ratio of effort (placement in the cage, more difficult cleaning,...) to benefit (more eggs, insect welfare,...)?
44) Line 455. Either which or that.
45) Line 483-485. The studies you mention here: Did they start experiments with adults that were similarly old and within one day after emergence from the pupae? If not this could have a big impact on oviposition peak and period.
46) Table 6. Please provide time as in Figure 3. Add the meaning of M and F in the caption.
47) Line 495-501. As mentioned in a previous comment, do you have data for the longevity? If so, please add and discuss them. Name exactly the supplementary you are referring to in line 501. Unfortunately, I don’t have access to the supplementary files of this manuscript and can’t check them.
48) Line 502-504. Please give only one decimal.
49) Line 512. Which supplementary exactly?
Comments on the Quality of English Language
Quality of English language is fine.
Author Response
Dear Reviewer,
Thank you very much for your detailed and thorough comments. You caught a lot of typos and formatting errors that we did not, and for this the manuscript is greatly improved.
Regards,
N. B. Lemke
Specific comments:
1) Line 22. …day 1…day 6…
Response: Improved the clarity of this line.
2) Line 34. …biotic and abiotic factors
Response: Changed to abiotic
3) Line 36. detritivores or detrivores
Response: Changed to detritivores.
4) Lines 43-45. Also consider adding Nayak et al 2024 (https://www.mdpi.com/2077-0472/14/1/8), where such parameters are mentioned and discussed in detail.
Response: I added this reference, thanks!
5) Line 63. Please use the “ – “ symbol instead of “-“ for ranges throughout the manuscript.
Response: Addressed throughout manuscript.
6) Line 105. …ran on a 12:12 L:D photoperiod…
Response: Added 'photoperiod'
7) Lines 107-109. Why didn’t you control the temperature and humidity? Please provide the range of both in the M&M section.
Response: Requested info is already found in section 2.4, lines 151-156
8) Line 112. Add space between number and unit
Response: Addressed
9) Line 113. Why did you use Monstera leaves instead of smaller ones? Please clarify.
Response: [Added “M. deliciosa shaped leaves were selected because it is a neotropical plant that is sympatric with Hermetia illucens in their native ranges.” Line 117 specifically says that larger leaves were NOT used.]
10) Line 114 and 119. Add the city of the companies.
Response: This information is not known.
11) Table 1. Use an appropriate table heading. Moreover, m2 has to be superscript.
Response: Addressed.
12) Lines 151-156. Use spaces between number and unit (but not for % since no unit). I think one decimal is precise enough, please change accordingly. Since you used an “wal-in incubator”, why are the temperatures and humidities so different between the trials? Is this incubator located outside?
Response: Addressed. Requested info is stated in section 2.4. There was an abnormal climactic event.
13) Line 175. Add space before “h”.
Response: Space added.
14) How did you know that the mating peak was reached during the trial? Please clarify.
Response: Changed text to clarify this. Reference to more detailed methodology is included.
15) Lines 180-182. Please provide length information in the same way like before. Use the same number of decimals here.
Response: Addressed.
16) Lines 186-189. Did you also consider multiple oviposition events per female?
Response: We considered that females can make multiple attempts, which was why number of clutches was also measured. It is also possible that females lay multiple clutches of eggs, but the methodology would have needed videography or some way of tracking females to determine if this was occurring.
17) Lines 191-193. Use the same font everywhere. One decimal is appropriate.
Response: Addressed.
18) Were the flies within the cages the same age (synchronized)? How old were they (days after emergence) at the beginning of the experiments? Did you exclude individuals that died within the first 1-2 days after the start of the experiment?
Response: This was a really good catch! I did not realize that the age range of the flies was missing. In trial 1 there weren’t any flies that died within the first two days. In trial 2, there were, and this is discussed in the supplementary info.
19) Line 190. Do you mean ml?
Response: Addressed.
20) Line 217-218. How large do you estimate the error that is introduced into the data by the unobserved time periods?
Response: While I didn’t record data for nighttime periods, I can say with confidence that there is little-to-no mating activity in the last hours of the day or in the early hours of the day, so the amount of information loss is negligible. Because I also counted the number of egg clutches, this would give an indication of how many oviposition events occurred during the unobserved time period as well. As for perching, I controlled for this in the time series analysis by taking a moving average of those unobserved periods.
21) Table 2. Please add that df means degrees of freedom. Use the same number of decimals for the respective metric described.
Response: Addressed.
22) Line 246. Mention that CI means confidence interval.
Response: Addressed.
23) Line 248. Use superscript for m3.
Response: Addressed.
24) Table 3. You mention once ** and once *** for the same value. Please revise accordingly. Please provide a more detailed table caption.
Response: Changed to *** .
25) Table 4. Change formatting of the left heading cell. Add a space between unit and number. Use the same number of decimals within the table.
Response: Addressed.
26) Line 259. Which supplementary exactly?
Response: Changed this reference to Table 5, since it is included in the main text already.
27) Lines 266-267. …towards male bias on day 2 and what?
Response: Addressed.
28) Table 5. Please highlight Trial 1 and 2 (better overview) and delete unnecessary lines in all tables.
Response: Addressed.
29) Figures 2 & 3. Please add axis lines and auxiliary lines to the scaling. Add axis labeling in Figure 3.
Response: I added primary horizontal gridlines to both figure 2 and 3.
30) Please make sure that there are spaces before and after a “=” symbol in the entire manuscript (figure captions, text, tables,…).
Response: Addressed.
31) Figure 5. Note the capitalization of the axis label (Observed, Trend, Seasonal, Random). Please add the unit “days” to Time. Moreover, you made a mistake while numbering the figures (figure 5 before figure 4), please change accordingly and check the text.
Response: Adressed.
32) Figure 4. Please change according to my comment 29). In the figure caption, please add “fly” after Black soldier. Why did you not filled in gaps here?
Response: Addressed. The gaps are not filled because quantitatively it doesn’t change the results.
33) Lines 303-307. One decimal is appropriate. Sometimes you write the trials capitalized, sometimes not. Please make sure to keep it the same throughout the whole manuscript. Also add exact supplementary file(s) in line 306.
Response: Addressed.
34) Line 309. Which declination? Since you started a new chapter, you should provide more information what you are writing about.
Response: Changed this to ‘decline’.
35) Lines 325-326. What would be the consequence for industrially insect farming?
Response: “To improve egg production per cage,” is already included here.
36) Line 346. Provide order and family.
Response: They are both Diptera: Culicidae - see next line.
37) Line 367. Add space after reference.
Response: Addressed.
38) Line 369. Delete the first “surface”.
Response: Addressed.
39) Line 374-378. Use superscript numbers.
Response: Addressed.
40) Line 385. Delete one of the “easily”.
Response: Addressed.
41) Did you found differences in longevity of the adults? Could this factor play a role?
Response: There was close to zero mortality in Trial 1. As for Trial 2, I analyzed factors contributing to differences between males and females Figures S11-S12 in the supplementary info. I also added a new table in the supplementary info (Table S1) that gives the sex-specific mortality.
Starting at line 297, I added the following: "Specifically, during Trial 2, 63.6 females died per cage by the end of the experiment on average, whereas just 26.6 males died per cage on average over the same period, representing a 2.39-fold difference (Table S1, Supplementary info)."
42) Line 394-405. What is actually known about (specific) volatiles that can be detected by the BSF adults? And what is known about color preferences?
Response: I added the following, which is the extent to which these are known.
"In addition, the effect of material (with different colors) for use as artificial perches has been investigated [54], showing increased performance when using a white material (19.4 ± 4.9 g eggs/cage) compared to green (15.0 ± 2.1 g eggs/cage) or a mix of the two (10.4 ± 1.5 g eggs/cage), although there was no significant difference between white and green."
43) Adding surface area (e.g. by artificial leaves) could therefore enable a higher density of flies/cage, which is promising. How do you evaluate the ratio of effort (placement in the cage, more difficult cleaning,...) to benefit (more eggs, insect welfare,...)?
Response: Without access to sensitive corporate information (amount of people working, their hourly pay rate, the number of cages, the time to clean, etc.), then this sounds like one would need to develop a simulation to estimate this trade-off. Hypothetically, one would need enough additional eggs produced to offset the cost of labor and the upfront material costs.
44) Line 455. Either which or that.
Response: Addressed.
45) Line 483-485. The studies you mention here: Did they start experiments with adults that were similarly old and within one day after emergence from the pupae? If not this could have a big impact on oviposition peak and period.
Response: Yes, the studies used similar aged flies.
46) Table 6. Please provide time as in Figure 3. Add the meaning of M and F in the caption.
Response: Addressed.
47) Line 495-501. As mentioned in a previous comment, do you have data for the longevity? If so, please add and discuss them. Name exactly the supplementary you are referring to in line 501. Unfortunately, I don’t have access to the supplementary files of this manuscript and can’t check them.
Response: This is referencing Figure S11-S12 in the Supplementary Info. I added Table S1 to the supplementary info as well.
48) Line 502-504. Please give only one decimal.
Response: Addressed.
49) Line 512. Which supplementary exactly?
Response: Addressed.